# Brief Communication: Sensitivity of Antarctic ice-shelf melting to ocean warming across basal melt models

Erwin Lambert[1*] and Clara Burgard[2,3*]

[1]Royal Netherlands Meteorological Institute (KNMI), Utrechtseweg 297, 3731 GA, De Bilt, The Netherlands
[2]Laboratoire d'Océanographie et du Climat: Expériments et Approches Numériques (LOCEAN), Sorbonne Université, CNRS/IRD/MNHN, Paris, France
[3]IGE, Univ. Grenoble Alpes, IRD/CNRS/INRAE/Grenoble INP, Grenoble, France
[*]These authors contributed equally to this work.

**Correspondence:** Erwin Lambert (erwin.lambert@knmi.nl)

**Abstract.** The uncertain sensitivity of Antarctic ice-shelf basal melt to ocean warming strongly contributes to uncertainties in sea-level projections. Here, we explore the response of five basal melt models to an idealised sub-thermocline warming. Total melt increases by 67%-240% (+ 1 °C) or by 141%-680% (+ 2 °C), showing a large intermodel spread. For deep regions of presently fast-melting ice shelves, this spread can reach two orders of magnitude. Therefore, a consistent calibration on present-day conditions does not guarantee consistent melt sensitivities and several basal melt forcings should be applied to prevent underestimating uncertainties in sea-level projections.

## 1 Introduction

The Antarctic Ice Sheet loses mass at an accelerating rate, leading to an increasing contribution to sea-level rise (Otosaka et al., 2023). This mass loss is mainly driven by increased ocean-induced melting at the base of Antarctic ice shelves (most recent estimates by Davison et al., 2023), leading to ice-shelf thinning, reduced buttressing and a consequential increase in the flux from the grounded ice toward the ocean. The future evolution of ocean-induced ice-shelf melt therefore plays a crucial role in projections of the Antarctic Ice Sheet evolution and its contribution to sea-level rise.

The most advanced way to simulate the ocean-induced ice-shelf melt, here referred to as basal melt, is to use a coupled ocean–ice-sheet model framework, where the ocean circulation in the cavities below the ice shelves is resolved and the ice-sheet geometry responds to the melt at the ice-shelf base. However, coupled ocean–ice-sheet models currently remain rare and computationally too expensive to run circum-Antarctic or global simulations at sufficiently high resolution over centennial to millenial time scales.

Instead, ice-sheet modelers rely on simpler models to provide basal melt forcing to Antarctic ice-sheet models (e.g. Jourdain et al., 2020). In recent decades, a variety of such dedicated basal melt models has been developed, ranging from simple parameterisations, assuming a linear or quadratic dependency on the difference between the ocean temperature and local freezing point (e.g. Jourdain et al., 2020), to simple models emulating the circulation in the cavity assuming plume dynamics (e.g. Lazeroms et al., 2018), an overturning circulation (Reese et al., 2018a), or both (Pelle et al., 2019), to more complex

models resolving the horizontal circulation component (Lambert et al., 2023) or relying on machine learning techniques (e.g. Burgard et al., 2023).

For their initial development, all basal melt models were calibrated and assessed on present-day observations or simulations close to present-day conditions and thus applied to force numerous ice-sheet models. However, the evaluation of a subset of models against a common reference revealed a large spread in the resulting melt rates and melt patterns (Burgard et al., 2022, 2023). An idealised ice-sheet modeling study also revealed the choice of basal melt model to dominate the uncertainty in mass loss and grounding line retreat (Berends et al., 2023). In addition, the application of several models on ocean properties

simulated by a coupled ocean–ice-sheet model showed large discrepancies in the melt resulting from ocean conditions warmer than present (Burgard et al., 2023), raising the question how the basal melt models differ in terms of their sensitivity to ocean warming.

    To assess the intermodel spread in basal melt sensitivity to ocean warming, we here compare a subset of five dedicated basal melt models of different levels of complexity. First, we calibrate the basal melt models based on an idealised ocean forcing

mimicking present-day conditions and on satellite-derived estimates of present-day melt. Second, we apply two different ocean forcings representing a sub-thermocline warming of $1\,°\,C$ and $2\,°\,C$, respectively, and quantify the change in melt rates and melt patterns. With this study, we highlight and interpret key differences in the behaviour of the various basal melt models for present-day and warm ocean conditions and provide an intermodel range of basal melt sensitivities for the 40 largest Antarctic ice shelves. As no observational dataset is available to evaluate the sensitivities, we cannot provide a recommendation of the

best or worst performing model. Instead, we aim to provide users with an overview of the behaviour of the different basal melt models in terms of patterns and sensitivity to ocean warming.

## 2   Data and Methods

To simulate a basal melt pattern, each model requires information on ice shelf cavity geometries as well as an ocean forcing in the form of a vertical profile of temperature and salinity in front of each ice shelf. In the following, we describe the input data

(Sec. 2.1), the five basal melt models (Sec. 2.2), the experimental setup (Sec. 2.3), and our calibration approach for the free parameters (Sec. 2.4).

### 2.1   Data

The ice-shelf geometries are based on the most recent version of the BedMachine dataset (BedMachine v3, Morlighem et al., 2020), which we regrid to a 2x2 km horizontal grid. We define the ice shelf domains based on masks from the Ice sheet Mass

Balance Intercomparison Exercise (IMBIE) community (Rignot et al., 2019). In order to allow the representation of meltwater exchange within shared cavities of neighbouring ice shelves, we merge ice shelves that have no clear distinction from an oceanic point of view (such as e.g. Filchner–Ronne and Crosson–Dotson) into one. Throughout this study, we focus on the 40 largest ice shelves of the remaining ensemble, containing all ice shelves with an area of at least 1000 $km^2$.

The observational estimates of basal melt used for the present-day calibration are taken from Paolo et al. (2023). These are
based on 26 years (1992–2017) of satellite altimetry. In this estimate, the total Antarctic integrated melt is 954 Gt yr$^{-1}$, which
is at the lower end of existing estimates, others ranging from 1080 Gt yr$^{-1}$ (Davison et al., 2023) to 1325 Gt yr$^{-1}$ (Rignot
et al., 2013) with uncertainties on the order of 200 Gt yr$^{-1}$. For calibration, we use the melt rates averaged over the whole
period.

## 2.2 Basal melt models

We compare the behaviour of five basal melt models: the Quadratic parameterisation, the PICO model, the Plume model, the
2D model LADDIE, and a Neural Network.

- In the Quadratic parameterisation (e.g. Jourdain et al., 2020), basal melting is expressed as a quadratic function of the
  difference between the local ocean temperature, directly extrapolated from the temperature in front of the ice shelf,
  and the local freezing point. Here, we adopt the quadratic-local formulation accounting for the local ice-shelf slope as
  described by Burgard et al. (2022).

- PICO (Reese et al., 2018a) is a box model that mimics an overturning circulation in the cavity, known as the ice-shelf
  pump (Lewis and Perkin, 1986). Only the bottom ocean properties in front of the ice shelf are applied at the grounding
  line and then modified while being advected along the ice-shelf draft, across a number of boxes dependent on the distance
  from the grounding line, leading to melting and freezing along the way.

- The Plume model (e.g. Lazeroms et al., 2018) is a one-dimensional model of a buoyant meltwater plume formed at the
  grounding line and rising along the ice-shelf draft following prescribed flowlines on the two-dimensional grid. We use
  the implementation by Burgard et al. (2022) but include an improved method to identify the origin grounding line depth
  by propagating plume paths from the grounding line to the rest of the ice shelf.

- The two-dimensional model LADDIE (Lambert et al., 2023) solves the vertically integrated Navier-Stokes equations in
  the upper mixed layer of the ice shelf cavity. Melt rates are computed using the commonly adopted "three-equations
  parameterisation" and the overturning is represented by a parameterised entrainment. The model can be seen as an
  extension of the one-dimensional Plume model, accounting for the horizontal flow under influence of geometric steering
  and Coriolis deflection.

- The Neural Network (Burgard et al., 2023) is a simple multilayer perceptron trained to emulate a cavity-resolving version
  of the ocean model NEMO at 1/4°horizontal resolution (more details about these simulations in Burgard et al., 2022).
  Compared to Burgard et al. (2023), the training dataset has been expanded with several projections, including those from
  Mathiot and Jourdain (2023), to account for warmer ocean conditions.

## 2.3 Experiments

For each model and ice shelf, three experiments are performed: a reference (REF) and two warming experiments (+1 °C and +2 °C). The REF experiment is designed to represent present-day conditions in an idealised framework. Ocean conditions are divided into six categories (Fig. 1a): Bellingshausen, East-Amundsen, West-Amundsen, Cool, Cold, and ISW (i.e. Ice Shelf Water). The first four contain a thermocline separating an upper layer at surface freezing temperature from a warm layer of Circumpolar Deep Water (CDW). The thermocline depth is taken at 500 m (similar to observations in the Amundsen Sea as in e.g. Dutrieux et al., 2014) except for the Bellingshausen Sea where a shallower thermocline exists; there, a depth of 380 m is taken.

The subsurface warm water mass has a temperature of 1.2 °C (Bellingshausen and East-Amundsen), 0.4 °C (West-Amundsen), and -1.2 °C (Cool). These values are inspired by in-situ observations (e.g. Dutrieux et al., 2014). The Cold ocean conditions are described by a full water column at surface freezing point. Finally, the ISW conditions contain a linear function of temperature of -0.3 °C per 1000 m to represent the presence of supercooled Ice Shelf Water, producing a temperature of approximately -2.1 °C at 1000 m, comparable to observations in the Filchner trough (Sallée et al., 2023). Salinities are taken such that a quadratic density stratification arises, again inspired by in-situ observations, except for ISW, where a linear stratification is imposed.

Several experimental choices, such as the Bellingshausen thermocline depth, the Cool sub-thermocline temperature, and the division of ice shelves between Cold and Cool forcing, could not be sufficiently constrained by observations. For these choices, to create plausible conditions, we apply a form of inversion in which we optimise the REF forcing using LADDIE to reproduce observed basal melt rates in the associated regions. As a consequence, the agreement between simulated melt rates by LADDIE and observational melt estimates may be slightly exaggerated in comparison to the other four models.

For each ice shelf, we restrict the intrusion of CDW into the cavity if there is a bathymetric obstacle at the ice-shelf front. For points within the cavity that are deeper than this bathymetric obstacle, temperature and salinity are extrapolated downwards from the deepest entrance depth.

For the two warming experiments, we add 1 °C and 2 °C, respectively, to the temperature below 500 m, which is the depth of our idealised CDW in most regions, and we leave the upper layer unchanged. The applied temperature profiles are visualised in Fig. 1a . A salinity anomaly is applied to compensate for the temperature anomaly, such that the density profile is identical for all experiments. By assuming a constant density across the experiments, we do not account for the increasing stratification which would result from the increased input of freshwater into the ocean. As shown by e.g. Lambert et al. (2024), this increased stratification can amplify the subsurface warming and thereby further increase the sensitivity of ice-shelf melt to ocean warming. By ignoring this feedback, our melt sensitivities may be conservative estimates.

From these three experiments, we extract three indicators describing the melt sensitivity of the ice shelves according to the five basal melt models in the range between reference ocean conditions and a 2° C warming. First, a 'melt sensitivity' is defined as:

$$\frac{dM}{dT}\big|_{(REF,+1°C)} = \frac{M_{+1°C} - M_{REF}}{T_{+1°C} - T_{REF}}. \tag{1}$$

Here, $M$ is the melt rate averaged over a specific region of an ice shelf. In this study, we use the melt averaged over the entire ice shelf $\bar{M}$ or over the deepest 10% of an ice shelf $M_{10}$. $T$ is the temperature forcing at the deepest point of this ice shelf. If the cavity is unblocked by bathymetric obstacles and the ice shelf base extends well below the thermocline depth, the temperature difference is 1 °C. Otherwise, the temperature difference is between 0 and 1 °C (Fig. 1a). The second indicator we introduce is the 'nonlinearity' of this sensitivity, which is defined as:

$$\frac{d^2\bar{M}}{dT^2} = \left(\frac{d\bar{M}}{dT}\Big|_{(+1°C,+2°C)} - \frac{d\bar{M}}{dT}\Big|_{(REF,+1°C)}\right)/(T_{+1.5°C} - T_{+0.5°C}). \tag{2}$$

This metric quantifies how the melt sensitivity changes when ocean temperatures increase. Positive values indicate that a +2 °C warming induces more than twice the melt increase of a +1 °C warming. Finally, changes in basal melt in the deep regions of ice shelves may have a larger impact than changes in shallower regions and melt sensitivities in deep regions may be considerably higher than ice-shelf averages (e.g. Jourdain et al., 2020; Reese et al., 2018b). Hence, as a third metric, we define the 'deep amplification' $DA$ of the melt sensitivities:

$$DA = \frac{dM_{10}}{dT}\Big|_{(REF,+1°C)}/\frac{d\bar{M}}{dT}\Big|_{(REF,+1°C)}. \tag{3}$$

Values above 1 indicate that the melt sensitivity in the deep region exceeds the ice-shelf average melt sensitivity.

## 2.4 Calibration

To enable a consistent intercomparison between the different basal melt models, we calibrate them on the same reference state, which mimicks present-day conditions. We use the temperature and salinity profiles from the REF experiment as forcing and calibrate the models to minimise the absolute difference to the observed ice-shelf melt (Paolo et al., 2023) averaged over the 40 ice shelves (0.60 m yr$^{-1}$). Note that melt rates throught this paper are defined as meters of water equivalent per year. For the Quadratic parameterisation, this leads to a dimensionless turbulent exchange coefficient parameter $K$ of $7.8\times10^{-5}$.

Both PICO and the Plume model have two tuning parameters and trial and error indicates that the empirical functions between the parameters derived by Burgard et al. (2022) cause an underestimation of the melt. Instead, for PICO, we probe a plausible range of effective turbulent temperature exchange velocities $\gamma_T^\star$ and select the one leading to the maximum melt when using the empirical function given by Burgard et al. (2022), linking the overturning coefficient $C$ to $\gamma_T^\star$. We then infer the corresponding $C$ needed to reach the target melt, resulting in a turbulent temperature exchange velocity $\gamma_T^\star$ of $0.85\times10^{-5}$ m s$^{-1}$ and an overturning coefficient $C$ of $7.4\times10^{6}$ m$^{6}$ s$^{-1}$ kg$^{-1}$. For the Plume model, we probe the plausible parameter space (shown in Fig. 9a of Burgard et al., 2022) and find a minimum absolute difference for a Stanton number of $3.2\times10^{-4}$ and an entrainment coefficient $E_0$ of $11.9\times10^{-2}$.

As the REF profiles were partly designed based on an inversion-like method with LADDIE, no recalibration of LADDIE is needed. The main tuning parameters are the top drag coefficient $C_{d,top}$ of 1.1 and the minimum layer thickness $D_{min}$ of 6.6 m (Lambert et al., 2023). For the Neural Network, the 40 data points are insufficient for a robust new training. Hence, we use the version where training has been extended to projections and use it as a NEMO emulator, accepting potential biases in the reference state.

# 3 Results and discussion

## 3.1 Reference melt patterns

Overall, the consistent calibration on the circum-Antarctic ice-shelf melt yields similar average melt rates, between 0.58 and 0.60 m yr$^{-1}$, for four out of five basal melt models (Fig. 1c-f). The Neural Network, which could not be recalibrated, produces slightly lower average melt rates of 0.46 m yr$^{-1}$ (Fig. 1g), which is likely the consequence of a warm bias in the training dataset. In addition to average melt rates, all basal melt models reproduce the comparably higher melt rates for the ice shelves in the Amundsen Sea and Totten/Moscow University ice shelves (#10-14 and 27 in Fig. 1a). This illustrates that the contrast between warm and cold ocean conditions is appropriately translated to a contrast between high and low melt rates by all models. A comparison of the melt rates between Amundsen Sea (observed: O(10 m yr$^{-1}$)) and Filchner–Ronne (observed: O(0.1 m yr$^{-1}$)) reveals that this contrast is well captured in the Quadratic, PICO, and LADDIE models, while the Plume model and the Neural Network underestimate this contrast by an order of magnitude (O(5 m yr$^{-1}$) vs O(0.5 m yr$^{-1}$) respectively).

In terms of patterns of melting and freezing, the Quadratic parameterisation stands out in several ways (Fig. 1c). First, the patterns are purely governed by the depth of the ice shelf draft and by the local slope. Second, the model does not reproduce refreezing as this requires a representation of advection of meltwater. The purely depth-dependent pattern and explicit slope-dependence combine into a strong deep amplification. For the Quadratic parameterisation, this value (6.7) is twice the value derived from observations (3.3).

The spatial patterns of PICO are strongly constrained by the box pattern (Fig. 1d). The inherent assumption of highest melt in the first box closest to the grounding line leads to wide bands of relatively uniform melting along the grounding lines and, in the case of cold cavities, relatively uniform refreezing in the center and close to the ice shelf fronts. As the melt rates are by design weakly dependent on depth and independent from the draft slope, the deep amplification is relatively low (2.1) compared to observations.

The melt patterns of the Plume model (Fig. 1e) display a stronger heterogeneity than the Quadratic and PICO models. In this model, the local slope angle relative to the imposed meltwater pathway induces small-scale variations in melting and refreezing. Similar to PICO, the transition from melting to refreezing in the cold ice shelves occurs further away from the grounding line than in the observations. However, the deep amplification is nearly identical to the observed value, indicating that the depth- and slope-dependence of melt rates are well represented by this model.

In comparison to the prescribed meltwater flowlines in the Plume model, the resolved meltwater circulation by LADDIE (Fig. 1f) allows for a number of additional features. In particular, the impact of Coriolis deflection leads to an asymmetry in melting and refreezing rates which are concentrated near the Western boundaries, in qualitative agreement with the observed patterns. In addition, the resolved circulation leads to a better positioning of the transition from melting to refreezing, closer to the grounding line, than PICO and the Plume model. The deep amplification (4.2) is higher than observed, possibly related to the imposed minimum layer thickness, which can lead to an overestimation of the heat transport to the deepest grounding lines. Detailed tuning of this parameter, as done by Lambert et al. (2023), can constrain melting in the deep regions.

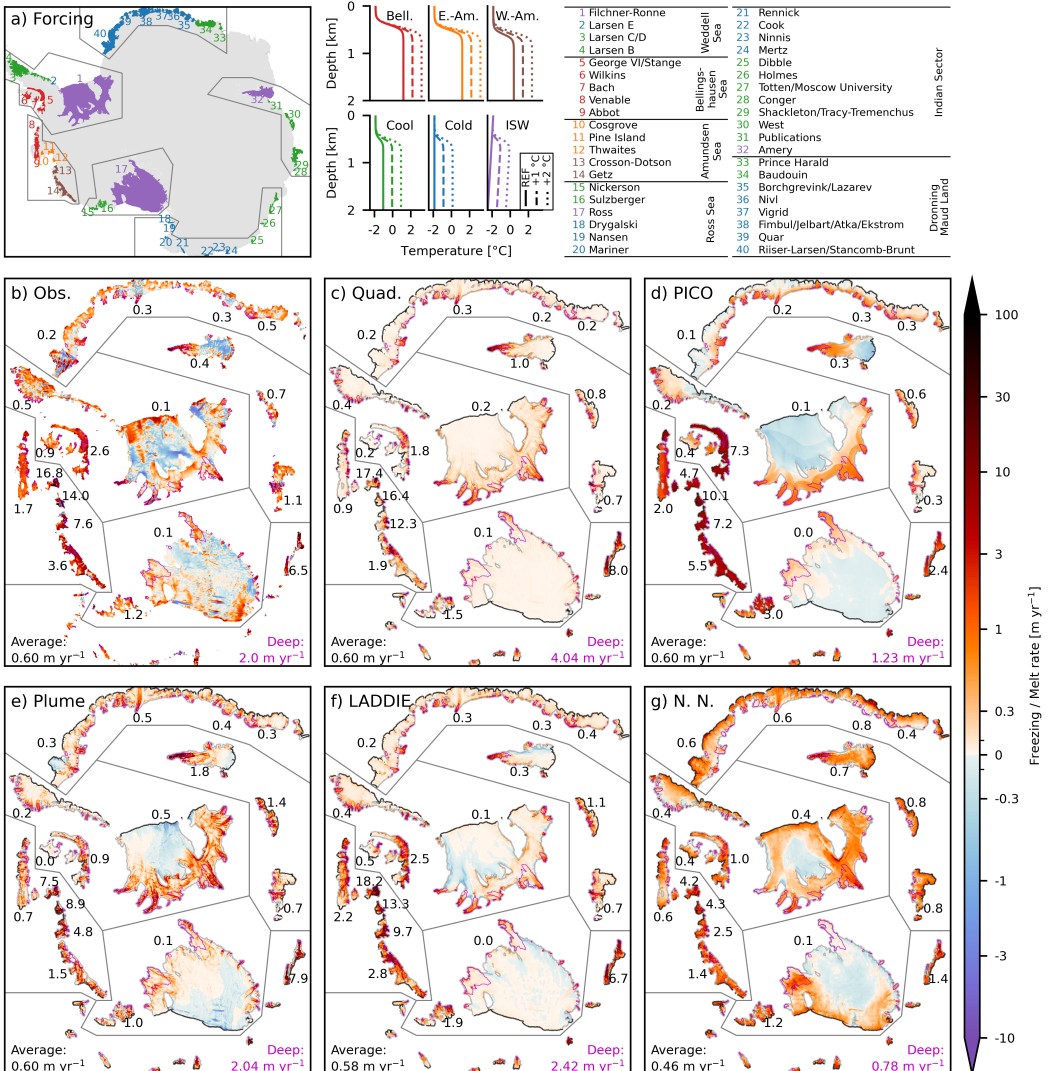

**Figure 1.** Ocean forcing and reference basal melt patterns. (a) Forcing, denoted by six colours, applied to 40 major ice shelves, and corresponding ice-shelf names. Temperature profiles applied in the reference (REF) and warming experiments. The map also displays the contours of the 'puzzle pieces' shown in panels (b) to (g). Basal melt patterns [m yr$^{-1}$] for (b) present-day observational estimates (Paolo et al., 2023) and (c) the Quadratic parameterisation, (d) PICO, (e) the Plume model, (f) LADDIE, and (g) the Neural Network (N.N.), forced with REF forcing. Grey (black) contours surrounding ice shelves denote the grounding line (calving front). The numbers shown along a selection of ice shelves are average melt rates in m yr$^{-1}$. The spatial average over the 40 ice shelves is shown in the bottom left; the average in the deepest 10% of each ice shelf (marked by magenta lines) is shown in the bottom right (magenta numbers).

The melting and freezing patterns simulated by the Neural Network (Fig. 1g) also reflect the Coriolis-deflected meltwater circulation as simulated by NEMO. This allows for a generally well-distributed pattern of melting and refreezing on the

Filchner–Ronne and Ross ice shelves. In addition, the Neural Network simulates 'mode 3' melting close to the ice shelf front, most visible along Dronning Maud Land and the Filchner–Ronne and Ross ice shelves. This melting is caused by the presence of Antarctic Summer Water (AASW), and cannot be explained by our forcing (hence this melting is not simulated by the other models). Instead, it is an intrinsic feature learned from the NEMO training dataset. The NEMO training dataset is based on annual averages, which intrinsically incorporate months in which AASW intrudes under the ice shelf, leading to melt near the front. Thus, the Neural Network will reproduce this pattern with any forcing. The Neural Network shows the lowest deep amplification (1.7), which is a reflection of lower melting at depth in NEMO due to a lower resolution and therefore a smoother and shallower ice draft.

## 3.2 Melt sensitivity

All ice shelves show an increase in basal melting in response to a sub-thermocline ocean warming by 1 °C (Fig. 2), ranging from 67% for the Neural Network to 240% for the Quadratic parameterisation, with a median of 175%. In all models, the melt sensitivity is stronger at depth, though the inter-model spread in sensitivities in the deep regions is an order of magnitude, ranging from 0.8 m yr$^{-1}$ (Neural Network) to 8 m yr$^{-1}$ (Quadratic). These sensitivities in the deep regions are a factor 1.7 (PICO) to 5.5 (Quadratic) higher than the ice-shelf average values. Average values of the melt rates for both +1 and +2 °C are summarised in Table S1 of the Supplementary Material, alongside Fig. S1, a reproduction of Fig. 2 for the +2 °C experiment.

The patterns of the melt response show distinct features related to the underlying model assumptions. The Quadratic parameterisation shows zero melt increase in the shallowest regions as it does not account for advection from the deeper regions where warming is applied. PICO shows a relatively uniform melt response which is strongly constrained by the box structure. The Plume shows a scattered melt response, mainly driven by the local slopes. LADDIE shows melt patterns reflecting the advective pathway of the simulated meltwater plumes. The Neural Network shows a melt response concentrated in specific regions, possibly reflecting a sensitivity in the cavity circulation in NEMO.

The melt sensitivities according to the different models can vary by an order of magnitude. In Fig. 3(a,b), we show the melt sensitivity $\frac{d\bar{M}}{dT}$ based on the +1 °C warming experiment. The unweighted distribution of the 40 ice shelves indicates that the Neural Network produces the lowest melt sensitivities, whereas the Quadratic parameterisation and LADDIE produce on average the highest sensitivities. Notably, both average and median sensitivities of all models fall below the uncertainty ranges of previous estimates of bulk sensitivities, derived from idealised modelling (LARMIP-2, Levermann et al., 2020) or from a calibration based on subsurface ocean warming and observed Antarctic ice discharge (van der Linden et al., 2023).

We have three explanations for this discrepancy. First, the models are calibrated on remote sensing estimates which are slightly lower than other estimates. This may result in a low bias in our modeled sensitivities. Second, as explained in Sec. 2.3, our experimental design with constant stratifications may further lead to a low bias in modeled sensitivities. An increased stratification would strengthen the buoyancy forcing in models that account for this (all except the Quadratic parameterisation); however, accounting for this melt–stratification feedback is a remaining challenge for stand-alone basal melt models. Third, as shown by Lambert et al. (2024), the calibration method of van der Linden et al. (2023) leads to an overestimation of melt

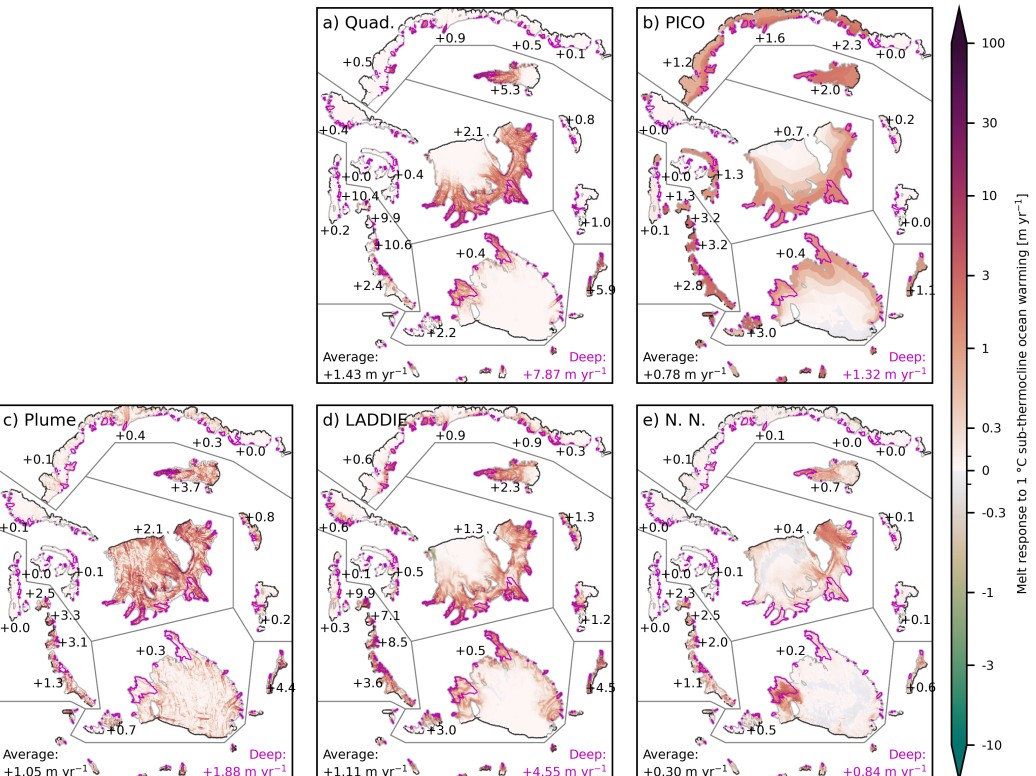

**Figure 2.** Spatial melt response to a 1 °C sub-thermocline warming expressed as the increase in melt relative to the reference (shown in Fig. 1) for (a) the Quadratic parameterisation, (b) PICO, (c) the Plume model, (d) LADDIE, and (e) the Neural Network (N.N.). Grey (black) contours surrounding ice shelves denote the grounding line (calving front). The numbers indicate the average response for selected ice shelves. The numbers at the bottom denote the Antarctic-wide average response (left) and the Antarctic-wide response averaged over the deepest 10% of all ice shelves (bottom right in magenta).

sensitivities when ignoring the positive feedback between meltwater and subsurface ocean warming. Taking into account these causes of discrepancy can lead to a better agreement between our melt sensitivities and other estimates.

On the individual ice-shelf level, some consensus on the melt sensitivity is found between the models, despite the overall strong intermodel disagreement. All models agree on a general classification of the major ice shelves, denoted by bold-faced names in Fig. 3, into three categories: those with comparably weak, moderate, or strong melt sensitivities. For the Larsen C/D (#3), George VI/Stange (5), Abbot (9), Ross (17), and Shackleton/Tracy Tremenchus (29) ice shelves, all models produce a weak sensitivity (below 1.5 m yr$^{-1}$ °C$^{-1}$) of the ice-shelf average melt. In addition, all models agree to some extent on a moderate sensitivity for the Filchner–Ronne (1), Getz (14) and Amery (32) ice shelves. Although the current literature provides limited material for comparison, Reese et al. (2023) derived melt sensitivities for Filchner–Ronne ranging from 0.7 to 1.5 m yr$^{-1}$ °C$^{-1}$) based on three-dimensional ocean simulations. Compared to these estimates, the melt sensitivity of LADDIE (1.3 m yr$^{-1}$ °C$^{-1}$) is closest, whereas that of PICO agrees with the lower bound of this estimate (0.7 m yr$^{-1}$ °C$^{-1}$).

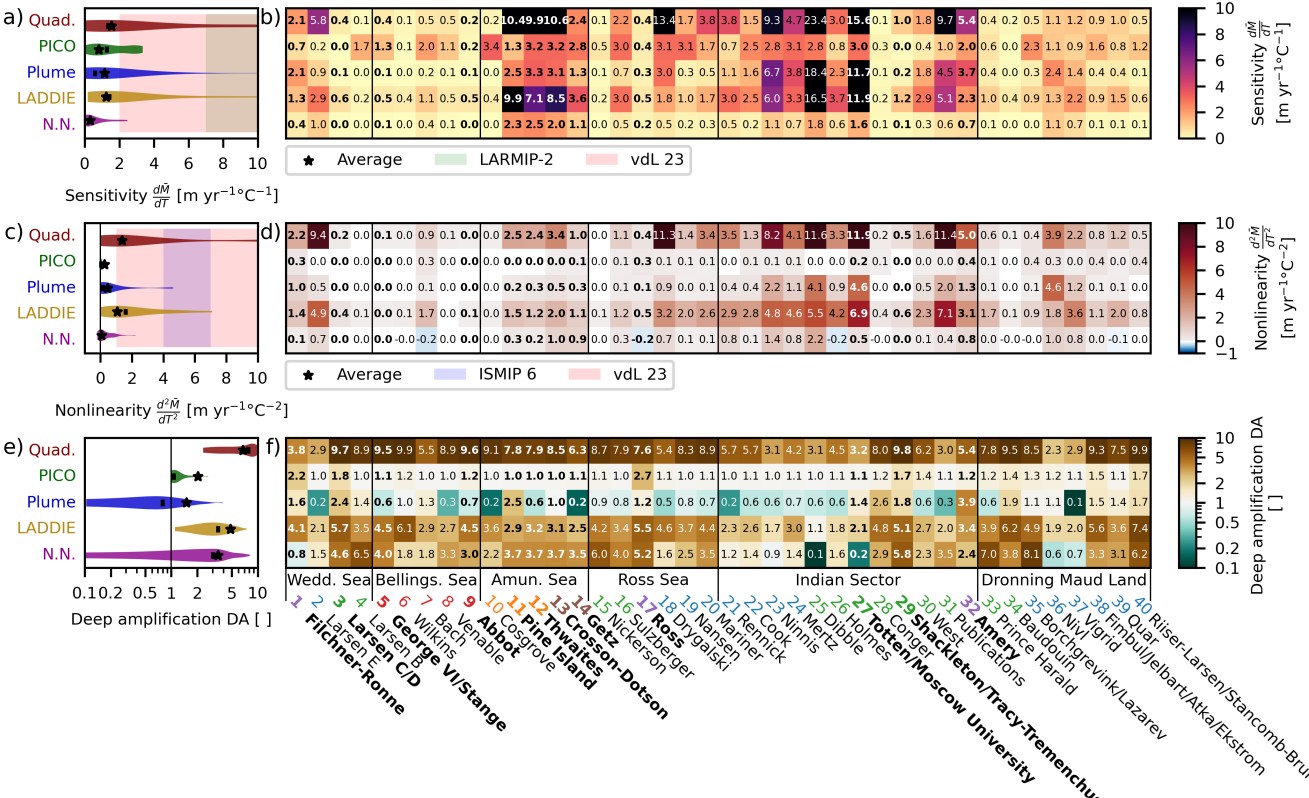

**Figure 3.** Melt sensitivity indicator distributions and values per ice shelf for each model. (a) Melt sensitivity distribution of all 40 ice shelves (unweighted). The star denotes the area-weighted mean and the vertical line is the median. Background shading are reference values from the literature: LARMIP-2 (full range, Levermann et al., 2020) and vdL 23 (5–95%, van der Linden et al., 2023). (b) Melt sensitivities averaged over individual ice shelves. (c) Distribution (unweighted) of the nonlinearity of melt sensitivities with shading displaying the nonlinearities from vdL 23 and from ISMIP6 (5–95%, Jourdain et al., 2020). (d) The nonlinearity of the average melt sensitivities for all individual ice shelves. A positive nonlinearity indicates that a +2 °C warming induces more than twice the melt increase of a +1 °C warming. (e) Deep amplification of the melt sensitivity (ratio deepest 10% versus ice shelf average) distribution across ice shelves and (f) values for each individual ice shelf. The value 1 indicates no deep amplification. The bold-faced names are the major ice shelves mentioned in the text, the coloured numbers are as in Fig. 1.

The third class on which all models qualitatively agree, contains the presently fast-melting ice shelves which display above-average melt sensitivities. These are the ice shelves located in the Amundsen Sea except Getz ice shelf (#10-13) and the Totten/Moscow University ice shelf (27). Quantitatively, however, the intermodel spread is considerably large. For the Crosson–Dotson, Thwaites, and Pine Island ice shelves in the Amundsen Sea, the melt sensitivities of the Quadratic parameterisation and LADDIE are approximately a factor 3 higher than those of the other three models (PICO, Plume, and Neural Network).

Again, Reese et al. (2023) provide comparison melt sensitivities based on in-situ observations at the fast-melting Dotson ice shelf ranging from 11 to 20 m yr$^{-1}$ °C$^{-1}$. The high sensitivities of Quadratic and LADDIE agree best with these estimates, indicating that the melt sensitivity in the Amundsen Sea may be understimated by the latter three models. For Totten/Moscow University, the sensitivities of the Quadratic paramaterisation, the Plume, and LADDIE are approximately a factor 5 to 6 higher than those of PICO and the Neural Network. More generally, the melt sensitivity of these presently fast-melting ice shelves is most dependent on the choice of the basal melt model.

To first order, the nonlinearity is correlated with the melt sensitivity (Fig. 3b,d). Both sensitivity and nonlinearity are strongest for LADDIE and the Quadratic parameterisation and weakest for PICO and the Neural Network. Notably, PICO shows a perfectly linear melt sensitivity (zero nonlinearity) for approximately half of the ice shelves; with a marked exception for the ISW ice shelves (Filchner–Ronne (1), Ross (17), and Amery (32)). For the Neural Network, 4 out of 40 ice shelves show a negative nonlinearity, reflecting a decrease in the melt sensitivity when the ocean warms up; this subset includes the Ross ice shelf (17). In comparison to previous estimates from the calibration of the Quadratic parameterisation to reference melt rates (Jourdain et al., 2020, ISMIP 6) or the calibration of a quadratic relationship between ocean temperatures and ice discharge (van der Linden et al., 2023), the nonlinearities in our estimates are relatively low. Based on the clear correlation between melt sensitivities (Fig. 3b) and nonlinearities (Fig. 3d), the causes for this discrepancy are likely the same as the causes for the discrepancy in melt sensitivities which were suggested previously. In addition, we find that the nonlinearity in the Quadratic parameterisation is strongest among the five models, indicating that the calibration by Jourdain et al. (2020) may lead to strong nonlinearity.

Considering the ice-shelf specific nonlinearities (Fig. 3d), again some consensus arises. All models agree that the melt sensitivities of George VI/Stange (5) and Abbot (9) are perfectly linear. Similarly, all models agree on a weak nonlinearity in the melt sensitivities of Larsen C/D (3), Ross (17), and Shackleton/Tracy-Tremenchus (29). In contrast, for all models, the nonlinearity of Totten/Moscow University (27) ranks among the highest values. Estimates of the nonlinearity can be derived from the melt sensitivities reported by Reese et al. (2023). For Filchner–Ronne, this is approximately 0.4 m yr$^{-1}$ °C$^{-2}$, comparable to PICO, but substantially lower than the estimates by the Plume, LADDIE, and the Quadratic parameterisation. In contrast, the observation-based estimate of the nonlinearity at the Dotson ice shelf (approximately 4.5 m yr$^{-1}$ °C$^{-2}$) is similar to the estimate by the Quadratic parameterisation, but substantially higher than all other models. Again, we suggest that the underestimation of an increase in stratification in warmer conditions may contribute to this discrepancy.

The deep amplification of melt sensitivities shows a particularly large intermodel spread which exceeds the spread between individual ice shelves (Fig. 3e-f). The Quadratic parameterisation gives the highest values with amplifications above a factor 5 for 30 out of 40 ice shelves. This high amplification can be attributed to the strong depth- and slope-dependence and the omission of meltwater advection, similar to the discrepancy between deep and shallower regions in the reference melt rates. PICO provides deep amplifications at the lower end, with a negligible amplification (values near 1) for most ice shelves with the notable exceptions being Filchner–Ronne and Ross. These low amplifications of PICO result from the relatively uniform melt patterns and melt sensitivities. The deep amplification of the Plume is lowest, with a majority of ice shelves giving a suppression of the melt sensitivity (values below 1) at depth. Finally, LADDIE and the Neural Network show a relatively good

agreement in deep amplifications for most ice shelves, with mid-range values around 2 to 5 for most ice shelves. The Neural Network, however, gives a melt suppression at depth for some individual ice shelves, which may be related to the fact that deep, shallow and narrow cavities and channels may not be resolved well in NEMO and therefore have been absent from the training dataset.

The large intermodel spread in both melt sensitivity and deep amplification results into extreme differences in terms of melt sensitivities in the deep regions. For example, the melt sensitivity in the deepest 10% of the Pine Island ice shelf ranges from 1.3 m yr$^{-1}$ °C$^{-1}$ (PICO) to 81 m yr$^{-1}$ °C$^{-1}$ (Quadratic). For Totten/Moscow University, this sensitivity in the deep regions ranges from 0.3 m yr$^{-1}$ °C$^{-1}$ (Neural Network) to 50 m yr$^{-1}$ °C$^{-1}$ (Quadratic). As melt in the deep regions most strongly impacts buttressing, we identify this intermodel spread in deep melt sensitivities - which can exceed two orders of magnitude in presently fast-melting ice shelves - as one critical source of model uncertainty in basal melt forcing. Further unravelling this intermodel spread should be a primary research focus to put into context the uncertainties in future projections of ice-sheet mass loss and sea-level rise and ultimately reduce them. The quantitative comparison of ice shelf-specific melt sensitivities and derivatives thereof, based on five dedicated melt models, can function as a starting point to address this issue.

## 4   Conclusions

For five basal melt models of varying complexity, an idealised sub-thermocline 1 °C warming scenario induces an increase by 67 to 240% in Antarctic-wide basal melting, depending on the model, with a median increase of 175%. For a 2 °C warming, the increase range from 141 to 680%, with a median of 415%. This large intermodel spread highlights the challenge to reduce uncertainties induced by basal melt in projections of Antarctic ice mass loss. We calibrated the five models to simulate a common reference state, resulting in comparable average melt rates for present-like conditions. However, already in these reference conditions, different model assumptions lead to large differences in the melt patterns. The intermodel disagreement is enhanced in the melt sensitivities to ocean warming, both for patterns and integrated metrics. Hence, a consistent calibration on present-day melt rates is no guarantee for a consistency in melt sensitivities. The intermodel disagreement applies in particular to melt sensitivities of individual ice shelves and their spatial patterns.

Within the assessed ensemble of five basal melt models, the Quadratic parameterisation has the highest melt sensitivity, strongest nonlinearity in the sensitivity and strongest amplification in the deeper regions. PICO displays an overall moderate melt sensitivity which is particularly uniform within each ice shelf and generally remains constant under further warming. The Plume model has an intermediate average melt sensitivity, which is suppressed (below-average) at depth for most ice shelves, and moderately nonlinear. LADDIE has an intermediate but strongly nonlinear melt sensitivity and an intermediate amplification at depth. Finally, the Neural Network has the lowest melt sensitivity, exhibiting a moderate nonlinearity, and an intermediate amplification at depth.

Our results can provide context to interpret more carefully simulations by ice-sheet models using one of these basal melt models. Each model has its advantages and disadvantages in terms of its ease of use, its ability to represent present-day melt patterns, and its potential realism of melt sensitivities to ocean warming. Hence, the models could not be ranked in terms

of overall quality, despite the large intermodel spread. Rather, these results can provide guidance for further development of the models or alternative calibration approaches. Moreover, we conclude that future ensemble-based studies – such as the next phase of the Ice-Sheet Model Intercomparison Project (ISMIP7) – should retain a diversity in their basal melt forcing to prevent underestimating uncertainties.

*Code and data availability.* Analysis code is available on https://github.com/erwinlambert/basal-melt-sens. The multimelt python package (https://github.com/ClimateClara/multimelt) was used for the Quadratic, PICO and Plume parameterisation. All code and data are available on Zenodo (Lambert and Burgard, 2024)

*Author contributions.* EL prepared the temperature and salinity profiles and performed simulations with LADDIE. CB prepared the input geometry and calibrated and performed simulations with the other four models. Both authors contributed equally to the analysis and writing.

*Competing interests.* The authors declare no competing interests.

*Acknowledgements.* The authors thank two anonymous reviewers for their constructive comments. The authors also thank N. Jourdain for the inspiration for the original approach to display the map of Antarctic ice shelves. A large part of the computations presented in this paper were performed using the GRICAD infrastructure (https://gricad.univ-grenoble-alpes.fr), which is supported by Grenoble research communities. EL was funded by the Netherlands Organization for Scientific Research (NWO) project HiRISE (grant no. OCENW.GROOT.2019.091). CB
was funded through the DEEP-MELT project (IRGA Pack IA 2021-2022), which is supported by MIAI @ Grenoble Alpes (ANR-19-P3IA-0003), through the European Union's Horizon 2020 research and innovation programme under grant agreement no. 821001 (SO-CHIC), through the European Union's Horizon Europe research and innovation programme under grant agreement no. 101081193 (OptimESM) and through the ANR under the France 2030 programme (ANR-22-EXTR-0008).

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
