# Peer review of "Brief Communication: Sensitivity of Antarctic ice-shelf melting to ocean warming across basal melt models"

_EGUsphere, 2024_

## Author Comment (AC5)

*RESPONSES TO REVIEWERS FOR*
*Brief Communication: Sensitivity of Antarctic ice-shelf melting to*
*ocean warming across basal melt models*
*E. Lambert\* & C. Burgard\*, submitted to The Cryosphere*

**Author response to reviewer #1**

The manuscript of Lambert and Burgard analyzes the sensitivity of five different basal ice shelf melting parameterizations to an idealized warming of 1°C. The five analyzed basal melting parameterizations are widely used or discussed and differ in complexity. The parameterizations range from point-dependent approaches (quadratic function of the temperature difference between the ocean and local freezing point temperature) to horizontal extensions of models representing different aspects of the ice shelf pump overturning circulation (PICO and Plume model), vertically integrated models solving the Navier-Stokes equation in the upper mixed layer within the ice shelf cavities, to a neural network trained by model output form the cavity-resolving simulations with the NEMO ocean model.

The authors drive these parameterizations with an observational-inspired hydrographical distribution of ocean temperature and salinity, which replicate the main hydrographic conditions for different regions and their ice shelf types (warm vs. cold water ice shelves, for instance), and they retune these parameterizations to reproduce contemporary observational basal melting distributions – except for the neural network due to the limit data base. Afterward, the authors applied a temperature increase of 1°C to analyze how the melting rates change under such a warming. The control run and the warming are analyzed regarding the changes in the overall basal melting rate averages and how melting increases in deeper (ocean depth) parts compared to the overall and shallower ice shelf areas.

In addition to the distinct differences in terms of basal melting amplification due to the warming and the related spatial signatures, all parameterizations show the highest melting amplifications for the warm-water ice shelves located in the Amundsen Sea Embayment, intermediate sensitivities for some ice shelf groups, and a weak reaction to the warming for large cold-water ice shelves (e.g., FRIS, RIS).

It was a pleasure to read the well-structured and prepared manuscript. The figures are of high quality, necessary, and informative.

Various groups are working on the future evolution of the Antarctic Ice Sheet (AIS), where ocean-driven mass loss predominates, and basal melting of floating ice shelves accounts for about 40–66% of the ocean-driven ice mass loss (Rignot et al. 2013; Depoorter et al. 2013; Liu et al. 2015; Davison et al. 2023). Hence, this study is an important contribution to understanding how different basal melting parameterizations drive future mass loss. Therefore, this work is also highly relevant for studies addressing Antarctica's sea level contribution in the coming centuries. Since the ocean-driven basal melting is central to the health of Antarctica, this work

**is also intriguing for the ice sheet modeling community and the coming Ice Sheet Modelling Intercomparison Project (ISMIP).**

**I recommend the publication of the manuscript after minor corrections.**

Thank you very much for the very positive feedback and for your constructive comments on how to further improve our manuscript. We agree with most suggestions and will incorporate these in the revision of our manuscript. In the following, we provide a point-by-point response.

*General comments*

**The manuscript is well organized and written.**

Thank you, we appreciate this positive feedback.

**Your manuscript addresses the basal melting enhancement for increased ocean temperatures. Since it is often discussed whether a particular parameterization shows a linear or quadratic behavior for increased temperatures, I wish you could perform an analysis for a temperature rise greater than 1°C, such as 0.5°C or 2°C, add the related results, and indicate if those different parameterizations have a linear or quadratic behavior.**

We agree on the added value of an additional scenario to assess the melt sensitivities. We will therefore include a +2 degree experiment as a third data point. In addition, based on your comments, those of the other reviewer, and our own assessment, we have concluded that a quadratic sensitivity is ambiguous and may invite different interpretations. Therefore, we propose to complement the linear sensitivity with a quantification of the nonlinearity instead. This nonlinearity is defined as the second-order derivative of the relation between melt and temperature, derived from the three data points (reference, +1 degree and +2 degrees). The benefit of this approach is that the metric can still be compared to quadratic sensitivities and additionally functions as an assessment of how valid the application of a linear sensitivity is to larger temperature perturbations. We hope that this solution satisfies the concerns of the reviewer.

**Have you considered including a linear parameterization in addition to the Quadratic parameterization? How would it behave compared to the other parameterizations listed in Section 2.2, Basal melt models (page 3)?**

Yes, we did consider the inclusion of the linear parameterisation. However, in the interest of space, we decided to restrict ourselves to a subset of the most widely used parameterisations and chose to exclude the linear parameterisation. Also, in Burgard et al. (2022), which assessed a range of parameterisations, the linear parameterisation performed worst.

**When it comes to the reference of the basal melting rate of (Paolo et al. 2023), I wish you could compare your reference with other estimates and how large the spread is because it would relate the found sensitivities of the analyzed parameterization to the uncertainty of current basal melting estimates, such as (Rignot et al. 2013; Depoorter et al. 2013; Liu et al. 2015; Davison et al. 2023).**

We have considered this suggestion. However, we think that a comprehensive discussion of the relationship between observational uncertainties and melt sensitivities is beyond the scope of this study, particularly in the context of the compact format of a brief communication. As we do agree that these uncertainties are relevant and significant, we will mention the spread in the total Antarctic melt between Paolo et al. 2015 and alternative datasets and briefly discuss the implications.

**You may recheck whether you use British or American English. I recognize mostly British English, but you use "e.g.," an American syntax. Please correct it.**

Thank you for pointing out this stylistic error. We will correct this in the manuscript.

**Specific comments**
*Main document*

**Line 9/L 9: You may add: "… loss is mainly driven by amplified ocean-induced melting … ."**
We will clarify this sentence by adding "increased" in front of ocean-induced melting.

**L 13: I'm unsure that "best" is meaningful here. You may rephrase it, e.g., " to as basal melt, is consistently simulated … ."**
We will reformulate.

**L 15: You may add: "… currently remain rare and computationally too expensive to run … ."**
Thank you for the suggestion. We will incorporate it.

**L 62: You may expend the sentence: "that mimics the overturning circulation in the cavity; known as ice-shelf pump (Lewis and Perkin 1986)".**
We will expand as suggested.

**L 72: Do you think the three-equation model is linear with respect to the temperature forcing? If so, please consider modifying the sentence "commonly adopted 'three-equations parameterization,' which is linear in the temperature forcing, and the overturning … ."**
The three-equations parameterisation, in our formulation, is not linear in temperature forcing. Temperature (and its gradients) impacts the horizontal velocities and thereby the friction velocity that appears in the turbulent exchange coefficients of both heat and salt. As we consider this discussion to be too detailed for the brief model description in the current manuscript, we will not elaborate on this further.

**L 84: You may add some information about CDW to address a wider audience, e.g., "… a warm layer of Circumpolar Deep Water (CDW), which a temperature of ≥0°C."**
Thank you for pointing this out. We will add this information.

**L 87: You may modify "The subsurface warm water mass … ."**
We will modify it.

**L 88: Please delete "where possible"**
Deleting "where possible" would be inconsistent, as not all values are directly inferred from observations. This is explained in the next paragraph. Hence, we will keep this sentence as is.

**L 90–91: I do not fully agree with the description of the water masses since the lowest temperature of HSSW corresponds to ocean water's surface freezing point temperature (about -1.87°C). In contrast, the water mass that is supercooled in relation to the surface freezing point temperature is Ice Shelf Water (ISW). The interaction of the HSSW with the ice shelf base transforms it into ISW. Please clarify this point.**
Thank you for pointing out this imprecision. We will rectify this.

**L 94: You may delete: "the exact values of"**
We will delete it.

**L 95: I am unsure, but should it be "… division of the ice shelves between Cold and Cool, … ."**
You are right, this should be "between". We will correct this.

**L 94–95: Since you create and use idealized ocean forcing, you may want to drop "cannot be sufficiently constrained by observations" since the idealization of observations is not necessarily identical. You may describe it like this: "Several experimental choices are made, such as … Cold and Cool case. Considering the idealized forcing, the value selection has a subjective component."**
Thank you for the suggestion. We will reformulate accordingly.

**L 99: You may replace the verb: "… ice shelf, we restrict CDW intrusion into … ."**
Thank you for the suggestion, we will replace it.

**L 109–112: Long sentence. You may consider splitting and rearranging it with the following sentence. For instance: "As changes … higher than ice-shelf averages (e.g., Jourdain et al., 2020). Hence, we additionally define the 'deep amplification,' where the nondimensional metric … ice-shelf average."**
We will follow your suggestion and restructure these sentences.

**L 122–124: It is unclear how the effective turbulent temperature exchange velocity is determined. Please clarify.**
Thank you for pointing this out, we will clarify.

**L 140 and L 141–142: Intriguing that the spreading factor is 100 = O(10 m yr-1)/O(0.1 m yr-1) in the first case and only 10 =O(5 m yr-1)/O(0.5 m yr-1) for the Plume Model and Neural Network.**

We agree that some differences in behaviour between the parameterisations are intriguing. As we could not discover a question or request in this comment, we will not adjust the manuscript in response to it.

**L 162–164: You speculate that the selected minimum layer thickness may overestimate the heat transport. Would a thicker or thinner layer thickness reduce the heat transport?**

This is not a trivial question and is beyond the scope of this paper to discuss. A thicker minimum thickness enhances entrainment and thus heat transport, a thinner layer thickness allows for a more efficient heat transfer to the ice shelf base. We will reformulate this sentence to provide some more insight.

**L 187 and 189: First, I was confused about what "linear sensitivities" and "quadradic sensitivities" mean. I guess you may something line "(T_cold – T_warm)/ Delta T" and "(T_cold – T warm)**2/Delta T ", or? Please clarify it.**

Thank you for pointing out that this was unclear. Partly in response to this comment, we have decided to redefine the sensitivities. In the revised manuscript, we will explicitly define the different metrics that appear in Fig 3.

**L 228: I am afraid I have to disagree that we can not avoid it, but it could be essential. Furthermore, some models/parameterizations may only be fit for some purposes. Therefore, reducing the intermodel spread might be misleading. Instead, knowing the limitations of the models and coming to a generalized description might be more critical.**

We note the concern and will reformulate accordingly.

**L 232: I am unsure about the style guide of Copernicus journals, but should each quantity have its own units so that it comes to "67% to 240%"?**

This is something we will double-check.

**L 197: You may be more implicit with your message: "... by the quadratic sensitivities (Fig. 3c), having on average the highest sensitivity, with values ... ."**

This appears to be an incorrect interpretation by the reviewer. The consensus relates to a weak sensitivity as mentioned in the previous sentence. We will therefore not adopt this suggestion.

**L 240: I am unsure if you would like to extend the sentence: "... melt enhancement in the deeper regions and none towards lowest depths: PICO ... ."**

We will reformulate as suggested.

*Figure*
**Figures 1 b—g) and 2 a—e): Great figures and a very smart way to use the available space to plot the ice shelf regions around Antarctica. Since your color bars have a "whitish" color around zero (0), it is not always clear what values are along the ice**

**shelf edge facing the ocean. Would it help to color the ocean (e.g., gray) and, therefore, mark the ice shelf edge?**

Thank you for pointing this difficulty out. Filling the ocean is not trivial with the way we constructed these figures. Hence, to address this issue, we will explore ways to highlight the grounding line and/or calving front.

**Bibliography**

Davison, Benjamin J., Anna E. Hogg, Noel Gourmelen, Livia Jakob, Jan Wuite, Thomas Nagler, Chad A. Greene, Julia Andreasen, and Marcus E. Engdahl. 2023. "Annual Mass Budget of Antarctic Ice Shelves from 1997 to 2021." *Science Advances* 9 (41): 1–12. https://doi.org/10.1126/sciadv.adi0186.

Depoorter, M.A, J.L. Bamber, J.A. Griggs, J.T.M. Lenaerts, S.R.M. Ligtenberg, M.R. van den Broeke, and G. Moholdt. 2013. "Calving Fluxes and Basal Melt Rates of Antarctic Ice Shelves." *Nature* 502 (7469): 89–92. https://doi.org/10.1038/nature12567.

Lewis, E. L., and R. G. Perkin. 1986. "Ice Pumps and Their Rates." *Journal of Geophysical Research: Oceans* 91 (C10): 11756–62. https://doi.org/10.1029/JC091iC10p11756.

Liu, Yan, John C Moore, Xiao Cheng, Rupert M Gladstone, Jeremy N Bassis, Hongxing Liu, Jiahong Wen, and Fengming Hui. 2015. "Ocean-Driven Thinning Enhances Iceberg Calving and Retreat of Antarctic Ice Shelves." *Proceedings of the National Academy of Sciences* 112 (11): 3263–68. https://doi.org/10.1073/pnas.1415137112.

Paolo, Fernando S., Alex S. Gardner, Chad A. Greene, Johan Nilsson, Michael P. Schodlok, Nicole-Jeanne Schlegel, and Helen A. Fricker. 2023. "Widespread Slowdown in Thinning Rates of West Antarctic Ice Shelves." *The Cryosphere* 17 (8): 3409–33. https://doi.org/10.5194/tc-17-3409-2023.

Rignot, E., S. Jacobs, J. Mouginot, and B. Scheuchl. 2013. "Ice-Shelf Melting Around Antarctica." *Science* 341 (6143): 266–70. https://doi.org/10.1126/science.1235798.

---

## Author Comment (AC6)

*RESPONSES TO REVIEWERS FOR*
*Brief Communication: Sensitivity of Antarctic ice-shelf melting to ocean warming across basal melt models*
*E. Lambert\* & C. Burgard\*, submitted to The Cryosphere*

Author response to reviewer #2

*Summary*
**This paper applies five approaches to ice shelf basal melt modeling to the 40 largest Antarctic ice shelves in order to compare their sensitivity to an idealized ocean warming scenario. These include more established approaches such as a simple parameterization of pointwise melt rates based on the regional hydrography and local ice base slope and more complex parameterizations accounting for meltwater advection and refreezing, and a newer machine learning approach using a neural network. The neural network is trained on output from a 1/4º NEMO simulation, which employs the same three-equation melt parameterization used in the intermediate complexity models. The aim of the neural network approach is to capture more of the complex spatial structure of basal melt rates without the prohibitively high computational cost of the full NEMO simulation.**

**The ocean conditions of the 40 ice shelves are classified into 6 categories based on their deep and near-surface temperatures and thermocline depths, each with an idealized "reference" temperature and salinity profile. Simulations are run for each ice shelf with its reference hydrography and then with a (salinity-compensated) 1ºC warming applied to the deep waters. The resulting melt rate distributions for the reference simulations are compared to observed melt rates to evaluate the fidelity of each modeling approach, and the warm simulations are compared with the reference to evaluate the sensitivity to warming.**

**The results show large differences in the spatial distribution of melt rates, even in the reference simulations which were calibrated to have the same average magnitude. The increases in the warming scenario vary among the models in both magnitude and distribution. However the models generally agree that the fastest-melting ice shelves under present day conditions are also most sensitive to warming.**

**Characterizing these differences is useful to the Antarctic Ice Sheet/Ocean modeling community. I recommend this paper to be published with some revisions.**

Thank you very much for underlining the significance of our work and for your insightful comments. We agree with most suggestions and will incorporate these in the revision of our manuscript. In the following, we provide a point-by-point response.

**General comments**
**The paper is well-structured and concise, with the figures in particular accommodating a huge amount of information in an impressively small package.**

Thank you for this positive feedback. We appreciate it.

**My major question is related to the sensitivity calculations and comparison with other studies. Because there was only one warming experiment, the quadratic sensitivities were calculated using the additional point of zero melt at the freezing point. Is this a common approach to calculating climate sensitivity? Would it be possible/valuable to conduct an additional simulation with a larger forcing in order to better constrain both the linear and quadratic sensitivities? If feasible, this could also allow one to evaluate whether the sensitivity of each model (and/or ice shelf system) is better characterized as linear or quadratic.**

We agree that the zero point appears somewhat arbitrary and will include, as suggested, an additional simulation with a +2 degree warming. The omission of this zero point naturally changes the meaning of the quadratic sensitivity, as it no longer sensibly applies to temperature forcings outside our explored range. Hence, we replace this quadratic sensitivity with a 'nonlinearity', which is defined as the second-order derivative over the explored range. The benefit is that this metric can still be compared to previous estimates of the quadratic sensitivity. In addition, the nonlinearity functions as an assessment of how valid it is to apply a linear sensitivity to larger temperature perturbations. We hope this solution satisfies the reviewer.

**Relatedly, where the sensitivities calculated in this study are compared to published estimates (e.g. lines 187-192), it would be helpful to include at least a brief description of the approaches of those studies and how they differ from the present work. Certainly the reader can visit those references for more detail but I think that a bit of context within the text would be helpful and appropriate. There are a few other points where I think more discussion would be appropriate which I have highlighted in the line-by-line comments.**
Thank you for pointing it out. We will provide additional context to these studies.

*Line-by-line comments*
**Abstract: "diversity in basal melt forcing is presently unavoidable to prevent underestimating uncertainties in future projections." This statement is confusing to me because it's separated from the initial mention of sea level rise and also I'm not sure what you mean by "unavoidable." I would say something like "a range of basal melt forcings should be applied to incorporate this uncertainty in future projections of sea level rise."**
We will reformulate to clarify.

**Line 138: It might be helpful to refer here to the ice shelf label numbers, i.e. "(10-14 and 27 in Figure 1a)".**
Good point, we will add this information.

**Line 141: Does this imply that the contrast is reproduced better in the other models? Please clarify.**
Yes, the large contrast is better in line with observations, so here we can conclude that there is an actual underestimation by these two models. We will clarify this.

**Line 168-170: This has me a little confused about how the Neural Network approach works. I guess it is trained on simulations that include seasonality, but when given an ocean temperature profile modeled on winter conditions, the resulting melt pattern effectively represents an annual mean — is this true? (It doesn't seem like this has much impact on the melt sensitivity calculation since it looks like a lot of that signal cancels out, at least looking at Filchner-Ronne and Ross.)**
Yes, the Neural Network has been trained on yearly-averaged melt rates, therefore implicitly reproducing seasonality in these particular cases. We will add one or two sentences to further clarify where this comes from and underline that it should not impact our main conclusions.

**Line 177: At times I found it slightly confusing that "deep amplification" can refer to either the actual melt rate or the melt rate response/anomaly — this is a place where I think it is a bit unclear and you could clarify by writing "Combining the average melt rate response and its deep amplification..."**
We agree and will use this term 'deep amplification' solely as representing the amplification in the linear melt sensitivities, as appearing in Fig. 3. Hence, we will express the melt rates and melt changes at depth (Fig. 1 and 2) in absolute terms to avoid confusion. Throughout the text, we will ensure that 'deep amplification' is used unambiguously.

**Line 187-192: Why do you think the sensitivities calculated in this study are so much lower than previous estimates? Are there key contrasts with the approaches taken in those papers that can help the reader interpret your findings?**
Yes, we have a preprint submitted (doi: 10.5194/egusphere-2024-2257) where we conclude that the discrepancy in sensitivities with van der Linden et al can be explained based on large-scale meltwater-ocean temperature feedbacks. We will refer to this preprint and briefly summarise the explanation.

**Line 195: To me this is a somewhat uncommon use of the word "consensual," I would omit it as you've already said earlier in the sentence that the models agree so I don't think it's necessary (or it could be replaced with "consistent").**
Thank you for spotting this. We will remove it.

**Line 196-199: Some patterns begin to emerge here but they weren't immediately obvious to me with the large number of names, not all of which were completely familiar. One simple thing that would make it easier to parse is to reverse the order that you list the ice shelves in the sentence so they are in the same order as they are shown in Figure 3. You could also consider noting in the text what ocean conditions apply to each ice shelf, or including the number of each ice shelf corresponding to the legend in Figure 1a to make it easier to refer back.**
As suggested, we will change the order in which we mention the ice shelves and include the numbers in the text. Where it does not affect readability, we will refer to the applied forcing as well.

**Line 203: except for Getz.**

You are right, we will add this information.

**Line 208-210: What was the method/approach used by the study you're comparing to, and is there a clear reason to think that result is more realistic?**
This study is based on a timeseries of observations in front of Dotson ice shelf and so we consider these to be a realistic guidance for the Dotson ice shelf and the neighboring ice shelves. We will elaborate on this briefly in the manuscript

**Line 220-229: From what you've shown, I don't think it's possible to "reduce the intermodel spread" in reference to this suite of models because they are fundamentally so different from one another.**
**Rather, if the goal is to improve sea level projections, it seems to me that it's important to prioritize the regions of the ice shelf that exert the greatest influence on ice sheet dynamics and consider which models seem most trustworthy in those settings. Thinking of results from Reese et al. (2018) showing the disproportionate sensitivity of upstream ice dynamics to thinning in narrow channels near the grounding line, it's concerning to me that the Neural Network is trained on a model that likely performs worst in those areas. (But maybe you disagree!) On the other hand, if the goal is to capture the change in spatial distribution of basal melt more broadly under warmer ocean conditions, the Neural Network may be a good choice. I know you are limited in how much you can say about which model is "better" but I think it could add to the value of this paper if you went a bit further into the discussion of the implications of your findings.**
We think that you are rightly pointing one of the main difficulties surrounding basal melt calibration and evaluation. Different applications require different aspects of basal melt to be realistic. Our aim is to provide an objective evaluation of the melt sensitivity based on several (widely applied) dedicated melt models, so that (ice-sheet) modelers can make decisions based on their research question. We will expand the discussion to outline potential lessons different readers may draw from our conclusions.

**Figures 1 & 2: I think it would be helpful if you could add a coastline, or some shading to either the ocean or land, to help orient and delineate the ice shelves in the "puzzle" subplots.**
Thank you for pointing this out. We will explore ways to visualise the grounding line and/or calving front to aid the orientation.

additional reference:
Reese, R., Gudmundsson, G.H., Levermann, A. et al. The far reach of ice-shelf thinning in Antarctica. Nature Clim Change 8, 53–57 (2018). https://doi.org/10.1038/s41558-017-0020-x

---

## Author Response (AR1)

*RESPONSES TO REVIEWERS FOR*
***Brief Communication: Sensitivity of Antarctic ice-shelf melting to ocean warming across basal melt models***
*E. Lambert\* & C. Burgard\*, submitted to The Cryosphere*

We thank both reviewers for the in-depth review and insightful comments.

Below, we repeat our point-by-point reply to all review comments.
*In italics, we elaborate on the changes we have made in the manuscript.*

**Author response to reviewer #1**

***General comments***

**The manuscript is well organized and written.**

Thank you, we appreciate this positive feedback.

**Your manuscript addresses the basal melting enhancement for increased ocean temperatures. Since it is often discussed whether a particular parameterization shows a linear or quadratic behavior for increased temperatures, I wish you could perform an analysis for a temperature rise greater than 1°C, such as 0.5°C or 2°C, add the related results, and indicate if those different parameterizations have a linear or quadratic behavior.**

We agree on the added value of an additional scenario to assess the melt sensitivities. We therefore included a +2°C experiment as a third data point. In addition, based on your comments, those of the other reviewer, and our own assessment, we have concluded that a quadratic sensitivity is ambiguous and may invite different interpretations. Therefore, we decided to complement the linear melt sensitivity with a quantification of the nonlinearity instead. This nonlinearity is defined as the second-order derivative of the relation between melt and temperature, derived from the three data points (reference, +1°C and +2°C). The benefit of this approach is that the metric can still be compared to quadratic sensitivities and additionally functions as an assessment of the validity of applying a linear melt sensitivity is to larger temperature perturbations.

*We now include an explanation of the additional +2°C experiment and a more detailed explanation of the melt sensitivity and this nonlinearity in Sec. 2.3 and updated the analysis in Sec. 3.2 to account for the new definition of the nonlinearity.*

**Have you considered including a linear parameterization in addition to the Quadratic parameterization? How would it behave compared to the other parameterizations listed in Section 2.2, Basal melt models (page 3)?**

Yes, we did consider the inclusion of the linear parameterisation. However, in the interest of space, we decided to restrict ourselves to a subset of the most widely used parameterisations and chose to exclude the linear parameterisation. Also, in Burgard et al. (2022), which assessed a range of parameterisations, the linear parameterisation performed worst.

**When it comes to the reference of the basal melting rate of (Paolo et al. 2023), I wish you could compare your reference with other estimates and how large the spread is because it would relate the found sensitivities of the analyzed parameterization to the uncertainty of current basal melting estimates, such as (Rignot et al. 2013; Depoorter et al. 2013; Liu et al. 2015; Davison et al. 2023).**

We have considered this suggestion. However, we think that a comprehensive discussion of the relationship between observational uncertainties and melt sensitivities is beyond the scope of this study, particularly in the context of the compact format of a brief communication. Still, as we do agree that these uncertainties are relevant and significant, we now include the upper and lower limit of other observational estimates, based on Rignot et al. (2013), Depoorter et al. (2013), Adusumilli et al. (2020) and Davison et al. (2023) to put the Paolo et al. (2023) estimate into context.

*We included "In this estimate, the total Antarctic integrated melt is 954 Gt yr$^{-1}$, which is at the lower end of existing estimates, others ranging from 1080 Gt yr$^{-1}$ (Davison et al., 2023) to 1325 Gt yr$^{-1}$ (Rignot et al., 2013) with uncertainties on the order of 200 Gt yr$^{-1}$.". In addition, we reflect on the relatively low values from Paolo and its impact on our melt sensitivities in Sec. 3.2*

**You may recheck whether you use British or American English. I recognize mostly British English, but you use "e.g.," an American syntax. Please correct it.**

Thank you for pointing out this stylistic error. We corrected this in the manuscript.

**Specific comments**
*Main document*

**Line 9/L 9: You may add: "… loss is mainly driven by amplified ocean-induced melting … ."**

We clarified this sentence by adding "increased" in front of ocean-induced melting:
*"This mass loss is mainly driven by increased ocean-induced melting at the base of Antarctic ice shelves ..."*

**L 13: I'm unsure that "best" is meaningful here. You may rephrase it, e.g., " to as basal melt, is consistently simulated … ."**
We reformulated as follows:
*"The most advanced way to simulate the ocean-induced ice-shelf melt, here referred to as basal melt, ..."*

**L 15: You may add: "… currently remain rare and computationally too expensive to run … ."**
Thank you for the suggestion, we incorporated it:
*"However, coupled ocean--ice-sheet models currently remain rare and computationally too expensive to run circum-Antarctic or global simulations at sufficiently high resolution over centennial to millenial time scales."*

**L 62: You may expend the sentence: "that mimics the overturning circulation in the cavity; known as ice-shelf pump (Lewis and Perkin 1986)".**
We expanded as suggested:
*"PICO (Reese et al., 2018) is a box model that mimics an overturning circulation in the cavity, known as the ice-shelf pump (Lewis and Perkin, 1986)."*

**L 72: Do you think the three-equation model is linear with respect to the temperature forcing? If so, please consider modifying the sentence "commonly adopted 'thre-equations parameterization,' which is linear in the temperature forcing, and the overturning … ."**
The three-equations parameterisation, in our formulation, is not linear in temperature forcing. Temperature (and its gradients) impacts the horizontal velocities and thereby the friction velocity that appears in the turbulent exchange coefficients of both heat and salt. As we consider this discussion to be too detailed for the brief model description in the current manuscript, we did not elaborate on this further.

**L 84: You may add some information about CDW to address a wider audience, e.g., "… a warm layer of Circumpolar Deep Water (CDW), which a temperature of ≥0°C."**
Thank you for pointing this out. In hindsight, we realised this 0°C threshold does not apply to our 'Cool' forcing, so we omitted this suggestion.

**L 87: You may modify "The subsurface warm water mass … ."**
We modified it:

*"The subsurface warm water mass has a temperature of 1.2°C (Bellingshausen and East-Amundsen), 0.4°C (West-Amundsen), and -1.2°C (Cool)."*

**L 88: Please delete "where possible"**
Deleting "where possible" would be inconsistent, as not all values are directly inferred from observations. This is explained in the next paragraph. Hence, we kept this sentence as is.

**L 90–91: I do not fully agree with the description of the water masses since the lowest temperature of HSSW corresponds to ocean water's surface freezing point temperature (about -1.87°C). In contrast, the water mass that is supercooled in relation to the surface freezing point temperature is Ice Shelf Water (ISW). The interaction of the HSSW with the ice shelf base transforms it into ISW. Please clarify this point.**
Thank you for pointing out this imprecision.
*We rectified this in Figure 1a) and in the text:*
*"Finally, the ISW conditions contain a linear function of temperature of -0.3°C per 1000 m to represent the presence of supercooled Ice Shelf Water, producing a temperature of approximately -2.1°C at 1000 m, comparable to observations in the Filchner trough (Sallée et al., 2023)."*

**L 94: You may delete: "the exact values of"**
We deleted it.

**L 95: I am unsure, but should it be "… division of the ice shelves between Cold and Cool, … ."**
You are right, this should be "between". We corrected this:
*"Several experimental choices, such as the Bellingshausen thermocline depth, the Cool sub-thermocline temperature, and the division of ice shelves between Cold and Cool forcing, could not be sufficiently constrained by observations."*

**L 94–95: Since you create and use idealized ocean forcing, you may want to drop "cannot be sufficiently constrained by observations" since the idealization of observations is not necessarily identical. You may describe it like this: "Several experimental choices are made, such as … Cold and Cool case. Considering the idealized forcing, the value selection has a subjective component."**
Thank you for the suggestion.
We reformulated to further clarify that, when observations are not available, we use a method that results in plausible conditions that reduces the subjective component:

*"Several experimental choices, such as the Bellingshausen thermocline depth, the Cool sub-thermocline temperature, and the division of ice shelves between Cold and Cool forcing, could not be sufficiently constrained by observations. For these choices, to create plausible conditions, we apply a form of inversion in which we optimise the REF forcing using LADDIE to reproduce observed basal melt rates in the associated regions."*

**L 99: You may replace the verb: "… ice shelf, we restrict CDW intrusion into … ."**
Thank you for the suggestion, we replaced it:
*"For each ice shelf, we restrict the intrusion of CDW into the cavity if there is a bathymetric obstacle at the ice-shelf front."*

**L 109–112: Long sentence. You may consider splitting and rearranging it with the following sentence. For instance: "As changes … higher than ice-shelf averages (e.g., Jourdain et al., 2020). Hence, we additionally define the 'deep amplification,' where the nondimensional metric … ice-shelf average."**
We followed your suggestion and restructured these sentences as part of the restructuring of the definition of the metrics:
*"Finally, changes in basal melt in the deep regions of ice shelves may have a larger impact than changes in shallower regions and melt sensitivities in deep regions may be considerably higher than ice-shelf averages (e.g. Jourdain et al. 2020; Reese et al. 2018). Hence, as a third metric, we define the `deep amplification' DA of the melt sensitivities:*
*$DA = dM_{10}/dT|_{(REF,+1\degree C)} \; / \; dM/dT|_{(REF,+1\degree C)}$.*
*Values above 1 indicate that the melt sensitivity in the deep region exceeds the ice-shelf average melt sensitivity."*

**L 122–124: It is unclear how the effective turbulent temperature exchange velocity is determined. Please clarify.**
Thank you for pointing this out, we clarified it as follows:
*"Instead, for PICO, we probe a plausible range of effective turbulent temperature exchange velocities $\gamma_T^\star$ and select the one leading to the maximum melt when using the empirical function given by Burgard et al. (2022), linking the overturning coefficient C to $\gamma_T^\star$. We then infer the corresponding C needed to reach the target melt, resulting in $\gamma_T^\star=0.85 \times 10^{-5}$ and $C=7.4 \times 10^6$."*

**L 140 and L 141–142: Intriguing that the spreading factor is 100 = O(10 m yr-1)/O(0.1 m yr-1) in the first case and only 10 =O(5 m yr-1)/O(0.5 m yr-1) for the Plume Model and Neural Network.**

We agree that some differences in behaviour between the parameterisations are intriguing. As we could not discover a question or request in this comment, we did not adjust the manuscript in response to it.

**L 162–164: You speculate that the selected minimum layer thickness may overestimate the heat transport. Would a thicker or thinner layer thickness reduce the heat transport?**

This is not a trivial question and is beyond the scope of this paper to discuss. A thicker minimum thickness enhances entrainment and thus heat transport, a thinner layer thickness allows for a more efficient heat transfer to the ice shelf base.

We reformulated this sentence to provide some more insight:

*"Detailed tuning of this parameter, as done by Lambert et al. (2023), can constrain melting in the deep regions."*

**L 187 and 189: First, I was confused about what "linear sensitivities" and "quadradic sensitivities" mean. I guess you may something line "(T_cold – T_warm)/ Delta T" and "(T_cold – T warm)\*\*2/Delta T ", or? Please clarify it.**

Thank you for pointing out that this was unclear. Partly in response to this comment, we have decided to redefine the sensitivities.

*We now explicitly define the different metrics that appear in Fig 3, in Sec. 2.3.*

**L 228: I am afraid I have to disagree that we can not avoid it, but it could be essential. Furthermore, some models/parameterizations may only be fit for some purposes. Therefore, reducing the intermodel spread might be misleading. Instead, knowing the limitations of the models and coming to a generalized description might be more critical.**

We note the concern and reformulated accordingly:

*"As melt in the deep regions most strongly impacts buttressing, we identify this intermodel spread in deep melt sensitivities - which can exceed two orders of magnitude in presently fast-melting ice shelves - as one critical source of model uncertainty in basal melt forcing. Further unravelling this intermodel spread should be a primary research focus to put into context the uncertainties in future projections of ice-sheet mass loss and sea-level rise and ultimately reduce them. The quantitative comparison of ice shelf-specific melt sensitivities and derivatives thereof, based on five dedicated melt models, can function as a starting point to address this issue."*

**L 232: I am unsure about the style guide of Copernicus journals, but should each quantity have its own units so that it comes to "67% to 240%"?**

We did not find the style recommendation for this particular case and therefore rely on the copy-editing process to implement the right style.

**L 197: You may be more implicit with your message: "… by the quadratic sensitivities (Fig. 3c), having on average the highest sensitivity, with values … ."**
This appears to be an incorrect interpretation by the reviewer. The consensus relates to a weak sensitivity as mentioned in the previous sentence. In any case, we have now rewritten this part to interpret the nonlinearity metric.

**L 240: I am unsure if you would like to extend the sentence: "… melt enhancement in the deeper regions and none towards lowest depths: PICO … ."**
We reformulated this paragraph to include the insight from the nonlinearity metric. This particular sentence now reads:
*"PICO displays an overall moderate melt sensitivity which is particularly uniform within each ice shelf and generally remains constant under further warming."*

*Figure*
**Figures 1 b—g) and 2 a—e): Great figures and a very smart way to use the available space to plot the ice shelf regions around Antarctica. Since your color bars have a "whitish" color around zero (0), it is not always clear what values are along the ice shelf edge facing the ocean. Would it help to color the ocean (e.g., gray) and, therefore, mark the ice shelf edge?**
Thank you for pointing this difficulty out. Filling the ocean is not trivial with the way we constructed these figures.
*We now added fading contours for the grounding line and calving front in Fig. 1 and Fig. 2.*

**Author response to reviewer #2**

**General comments**
**The paper is well-structured and concise, with the figures in particular accommodating a huge amount of information in an impressively small package.**

Thank you for this positive feedback. We appreciate it.

**My major question is related to the sensitivity calculations and comparison with other studies. Because there was only one warming experiment, the quadratic sensitivities were calculated using the additional point of zero melt at the freezing point. Is this a common approach to calculating climate sensitivity? Would it be possible/valuable to conduct an additional simulation with a larger forcing in order to better constrain both the linear and quadratic sensitivities? If feasible, this could**

**also allow one to evaluate whether the sensitivity of each model (and/or ice shelf system) is better characterized as linear or quadratic.**

We agree that the zero point appears somewhat arbitrary and now include, as suggested, an additional simulation with a +2°C warming. The omission of this zero point naturally changes the meaning of the quadratic sensitivity, as it no longer sensibly applies to temperature forcings outside our explored range. Hence, we replace this quadratic sensitivity with a 'nonlinearity', which is defined as the second-order derivative over the explored range. The benefit is that this metric can still be compared to previous estimates of the quadratic sensitivity. In addition, the nonlinearity functions as an assessment of how valid it is to apply a linear melt sensitivity to larger temperature perturbations.

*We reformulated the definition of the metrics at the end of Sec. 2.3 and updated the analysis in Sec. 3.2 to account for the new definition of the nonlinearity.*

**Relatedly, where the sensitivities calculated in this study are compared to published estimates (e.g. lines 187-192), it would be helpful to include at least a brief description of the approaches of those studies and how they differ from the present work. Certainly the reader can visit those references for more detail but I think that a bit of context within the text would be helpful and appropriate.**

Thank you for pointing it out.

*Throughout Sec. 3.2, when we compare our results to previous studies, we now provide context on the approaches of these other studies.*

**There are a few other points where I think more discussion would be appropriate which I have highlighted in the line-by-line comments.**

*Line-by-line comments*
**Abstract: "diversity in basal melt forcing is presently unavoidable to prevent underestimating uncertainties in future projections." This statement is confusing to me because it's separated from the initial mention of sea level rise and also I'm not sure what you mean by "unavoidable." I would say something like "a range of basal melt forcings should be applied to incorporate this uncertainty in future projections of sea level rise."**
We reformulated to clarify as follows:
*"We conclude that a consistent calibration on present-day conditions does not guarantee consistent melt sensitivities and that the inclusion of various basal melt forcings should be applied to prevent underestimating uncertainties in sea-level projections."*

**Line 138: It might be helpful to refer here to the ice shelf label numbers, i.e. "(10-14 and 27 in Figure 1a)".**

Good point, we added this information:

*"In addition to average melt rates, all basal melt models reproduce the comparably higher melt rates for the ice shelves in the Amundsen Sea and Totten/Moscow University ice shelves (\#10-14 and 27 in Fig.1a)"*

**Line 141: Does this imply that the contrast is reproduced better in the other models? Please clarify.**

Yes, the large contrast is better in line with observations, so here we can conclude that there is an actual underestimation by these two models.

*We clarified this as follows:*

*"A comparison of the melt rates between Amundsen Sea (observed: O(10 m yr$^{-1}$)) and Filchner--Ronne (observed: O(0.1 m yr$^{-1}$)) reveals that this contrast is well captured in the Quadratic, PICO, and LADDIE models, while the Plume model and the Neural Network underestimate this contrast by an order of magnitude (O(5 m yr$^{-1}$) vs O(0.5 m yr$^{-1}$) respectively)."*

**Line 168-170: This has me a little confused about how the Neural Network approach works. I guess it is trained on simulations that include seasonality, but when given an ocean temperature profile modeled on winter conditions, the resulting melt pattern effectively represents an annual mean — is this true? (It doesn't seem like this has much impact on the melt sensitivity calculation since it looks like a lot of that signal cancels out, at least looking at Filchner-Ronne and Ross.)**

Yes, the Neural Network has been trained on yearly-averaged melt rates, therefore implicitly reproducing seasonality in these particular cases.

*We clarified as follows:*

*"This melting is caused by the presence of Antarctic Summer Water (AASW), and cannot be explained by our forcing (hence this melting is not simulated by the other models). Instead, it is an intrinsic feature learned from the NEMO training dataset. The NEMO training dataset is based on annual averages, which intrinsically incorporate months in which AASW intrudes under the ice shelf, leading to melt near the front. Thus, the Neural Network will reproduce this pattern with any forcing."*

**Line 177: At times I found it slightly confusing that "deep amplification" can refer to either the actual melt rate or the melt rate response/anomaly — this is a place where I think it is a bit unclear and you could clarify by writing "Combining the average melt rate response and its deep amplification…"**

We agree and now use this term 'deep amplification' solely as representing the amplification in the linear melt sensitivities, as appearing in Fig. 3.

*Hence, we now express the melt rates and melt changes at depth (Fig. 1 and 2) in absolute terms to avoid confusion. Throughout the text, we now use 'deep amplification' to describe the amplification of the melt sensitivity at depth.*

**Line 187-192: Why do you think the sensitivities calculated in this study are so much lower than previous estimates? Are there key contrasts with the approaches taken in those papers that can help the reader interpret your findings?**

Yes, we have a preprint submitted (doi: 10.5194/egusphere-2024-2257) where we conclude that the discrepancy in sensitivities with van der Linden et al can be explained based on large-scale meltwater-ocean temperature feedbacks.

*We now refer to this preprint and briefly summarise the explanation:*

*"Third, as shown by Lambert et al. (2024), the calibration method of van der Linden (2023) leads to an overestimation of melt sensitivities when ignoring the positive feedback between meltwater and subsurface ocean warming. "*

**Line 195: To me this is a somewhat uncommon use of the word "consensual," I would omit it as you've already said earlier in the sentence that the models agree so I don't think it's necessary (or it could be replaced with "consistent").**

Thank you for spotting this. We removed it.

**Line 196-199: Some patterns begin to emerge here but they weren't immediately obvious to me with the large number of names, not all of which were completely familiar. One simple thing that would make it easier to parse is to reverse the order that you list the ice shelves in the sentence so they are in the same order as they are shown in Figure 3. You could also consider noting in the text what ocean conditions apply to each ice shelf, or including the number of each ice shelf corresponding to the legend in Figure 1a to make it easier to refer back.**

As suggested, we changed the order in which we mention the ice shelves and included the numbers in the text. We also added the ice-shelf numbers (and in the color of the corresponding forcing) in Fig. 3.

**Line 203: except for Getz.**

You are right, we added this information:

*"These are the ice shelves located in the Amundsen Sea except Getz ice shelf (\#10-13) ..."*

**Line 208-210: What was the method/approach used by the study you're comparing to, and is there a clear reason to think that result is more realistic?**

This study is based on a timeseries of observations in front of Dotson ice shelf and so we consider these to be a realistic guidance for the Dotson ice shelf and the neighboring ice shelves.

*We know introduce this approach as follows:*
*"Again, Reese et al. (2023) provide comparison melt sensitivities based on in-situ observations at the fast-melting Dotson ice shelf ranging from 11 to 20 m $yr^{-1}$ $°C^{-1}$."*

**Line 220-229: From what you've shown, I don't think it's possible to "reduce the intermodel spread" in reference to this suite of models because they are fundamentally so different from one another.**

**Rather, if the goal is to improve sea level projections, it seems to me that it's important to prioritize the regions of the ice shelf that exert the greatest influence on ice sheet dynamics and consider which models seem most trustworthy in those settings. Thinking of results from Reese et al. (2018) showing the disproportionate sensitivity of upstream ice dynamics to thinning in narrow channels near the grounding line, it's concerning to me that the Neural Network is trained on a model that likely performs worst in those areas. (But maybe you disagree!) On the other hand, if the goal is to capture the change in spatial distribution of basal melt more broadly under warmer ocean conditions, the Neural Network may be a good choice.**

**I know you are limited in how much you can say about which model is "better" but I think it could add to the value of this paper if you went a bit further into the discussion of the implications of your findings.**

We think that you are rightly pointing one of the main difficulties surrounding basal melt calibration and evaluation. Different applications require different aspects of basal melt to be realistic. Our aim is to provide an objective evaluation of the melt sensitivity based on several (widely applied) dedicated melt models, so that (ice-sheet) modelers can make decisions based on their research question.

*We expanded the take-home message in this section and put additional effort into interpreting the inter-model spread. However, we refrain from ranking the models in terms of performance as the sparsely available (largely model-based) reference values of melt sensitivities are insufficient to do this:*

*"As melt in the deep regions most strongly impacts buttressing, we identify this intermodel spread in deep melt sensitivities - which can exceed two orders of magnitude in presently fast-melting ice shelves - as one critical source of model uncertainty in basal melt forcing. Further unravelling this intermodel spread should be a primary research focus to put into context the uncertainties in future projections of ice-sheet mass loss and sea-level rise and ultimately reduce them. The quantitative comparison of ice shelf-specific melt sensitivities and derivatives thereof, based on five dedicated melt models, can function as a starting point to address this issue."*

**Figures 1 & 2: I think it would be helpful if you could add a coastline, or some shading to either the ocean or land, to help orient and delineate the ice shelves in the "puzzle" subplots.**

Thank you for pointing this out.

*We now added fading contours for the grounding line and calving front in Fig. 1 and Fig. 2.*

---

## Referee Report (RR1)

*Review of Lambert and Burgard: "Brief Communication: Sensitivity of Antarctic ice-shelf melting to ocean warming across basal melt models."*

*The Cryosphere Discussion, Paper: 10.5194/egusphere-2024-2358*

The revised manuscript of Lambert and Burgard analyses the sensitivity of five different basal ice shelf melting parameterisations to an idealised warming of 1 °C and 2 °C, while the five analysed and widely used basal melting parameterisations differ in complexity.

The authors of the revised manuscripts have taken up almost all raised issues and suggestions by the reviewers. However, I would like to suggest some last-minute changes.

**I recommend the publication of the manuscript after very minor corrections.**

**General comments**

The manuscript addresses the basal melting enhancement for increased ocean temperatures. In the revised manuscript, the authors have performed an additional simulation for a warming of 2 °C. Since the brief communication shall be concise, the authors may summarise very general information about the reference and the two warming scenarios of 1 °C or 2 °C in a simple table. What do the authors think about the following example? The information could be provided as supplementary material if space is an issue.

| Model | M (m yr$^{-1}$) | M$_{10}$ (m yr$^{-1}$) | DA | Scenario |
|---|---|---|---|---|
| Observation | 0.60 | 2.00 | 3.3 | Reference |
| Quad. | 0.60 | 2.04 | 6.7 | Reference |
| | 2.03 | ? | ? | +1 °C |
| | ? | ? | ? | +2 °C |
| PICO | 0.60 | 1.23 | 2.1 | Reference |
| | ? | ? | ? | +1 °C |
| | ? | ? | ? | +2 °C |
| Plume | 0.60 | 2.04 | 3.4 | Reference |
| | ? | ? | ? | +1 °C |
| | ? | ? | ? | +2 °C |
| Laddie | 0.58 | 2.42 | 4.2 | Reference |
| | ? | ? | ? | +1 °C |
| | ? | ? | ? | +2 °C |
| N. N. | 0.46 | 0.78 | 1.7 | Reference |
| | ? | ? | ? | +1 °C |
| | ? | ? | ? | +2 °C |

As a supplement material, you may provide the spatial melting response to the 2 °C warming, like Figure 2 of the 1 °C warming.

**Specific comments**

Main document

Line 2–3/L 2–3: The authors may mention both warming scenarios in the abstract, e.g., "to an idealised sub-thermocline 1 °C and 2 °C warming … ."

L 72: Please drop the "e.g." because you use a specific implementation. You may write: "We use the implementation by Burgard et al. (2022) but include an …. ."

L 15: You may add: "… currently remain rare and computationally too expensive to run … ."

L 133: What is the unit of the $K$ value? Please add the unit.

L 138: What is the unit of the turbulent heat exchange ($\gamma^*_T$) value? Please add the unit.

L 142: I am unsure about the style guide of Copernicus journals, but should there be a space between the number and the following unit?

L 165: If the provided numbers in Figure 1 are taken, the deep amplification factor for PICO is 2.1. Please double-check.

L 187: As before, the authors may also add here the factor by writing: "… the lowest deep amplification (1.7), which is a … ."

L 193:  Should it be: "These sensitivities have a factor between 1.7 (PICO) and 5.5 (Quadratic) higher than the ice-shelf average values."?

L 226: The authors may rephrase "Quantitatively, however, the intermodel spread is very large" to "Quantitatively, however, the intermodel spread is considerably large."

L 279.: The authors may add a sentence summarising the results for 2 °C: "For  2 °C, the increases range from xx % to xx %, with a mean of xx %."

---

## Author Response (AR2)

Dear editor, please find the reviews below, with our point-by-point reply in green.

**Reviewer 1:**

Thank you for your work to improve the manuscript. The additional experiment and new metric for nonlinearity of melt response make this study stronger and the updated figures are easier to interpret.

We thank the reviewer for their constructive comments and positive overall review. Please find below our reply to the remaining comments.

In this version, you refer at several points to a feedback between melt and stratification/buoyancy, which is not explicitly described nor is a reference provided. This first comes up in line 109-110, again in line 209-211, and also in line 254-255. Please explain what you mean and be specific about why you expect that the estimated melt sensitivities may be conservative as a result.

We agree that this process was insufficiently explained. The reference previously provided is Lambert et al (2024). We have now included that reference in line 109-110 and expanded the relation between this feedback and melt sensitivities as follows:
*"By assuming a constant density across the experiments, we do not account for the increasing stratification which would result from the increased input of freshwater into the ocean. As shown by e.g. Lambert et al. 2024, this increased stratification can amplify the subsurface warming and thereby further increase the sensitivity of ice-shelf melt to ocean warming. By ignoring this feedback, our melt sensitivities may be conservative estimates."*

Also note that the abstract still references a warming of 1°C while the new version of the manuscript includes two warming experiments.

Thank you for pointing this out, we have now additionally mentioned the 2°C

**Reviewer 2:**

The revised manuscript of Lambert and Burgard analyses the sensitivity of five different basal ice shelf melting parameterisations to an idealised warming of 1 °C and 2 °C, while the five analysed and widely used basal melting parameterisations differ in complexity. The authors of the revised manuscripts have taken up almost all raised issues and suggestions by the reviewers. However, I would like to suggest some last-minute changes. I recommend the publication of the manuscript after very minor corrections.

We thank the reviewer for this positive review and their constructive comments. Please find our point-by-point reply below.

General comments
The manuscript addresses the basal melting enhancement for increased ocean temperatures. In the revised manuscript, the authors have performed an additional simulation for a warming of 2 °C. Since the brief communication shall be concise, the authors may summarise very general information about the reference and the two warming scenarios of 1 °C or 2 °C in a simple table. What do the authors think about the following example (please see attached pdf file)? The information could be provided as supplementary material if space is an issue.

We agree that this table may contain useful quantitative information for some readers and have therefore adopted the reviewer's suggestion to add this table as supplementary material.

As a supplement material, you may provide the spatial melting response to the 2 °C warming, like Figure 2 of the 1 °C warming.

Similar to the table, we have followed the reviewer's suggestion and added this figure to the supplementary material.

Specific comments
Main document
Line 2–3/L 2–3: The authors may mention both warming scenarios in the abstract, e.g., "to an idealised sub-thermocline 1 °C and 2 °C warming ... ."

Thank you for pointing this out, we have now additionally mentioned the 2ºC

L 72: Please drop the "e.g." because you use a specific implementation. You may write: "We use the implementation by Burgard et al. (2022) but include an .... ."

Agreed and corrected

L 15: You may add: "... currently remain rare and computationally too expensive to run ... ."

This line was already there in the resubmitted version.

L 133: What is the unit of the K value? Please add the unit.

As defined in Burgard et al. 2022, K is a dimensionless parameter. We have now specified this explicitly in the text.

L 138: What is the unit of the turbulent heat exchange (γ*T) value? Please add the unit.

Thank you for pointing out that these units were missing. The unit for \gamma is m/s and for C it is m^6/s/kg. We added the units in the manuscript.

L 142: I am unsure about the style guide of Copernicus journals, but should there be a space between the number and the following unit?

Agreed, we have corrected this and made the inline presentation of calibrated parameters consistent throughout the paragraphs.

L 165: If the provided numbers in Figure 1 are taken, the deep amplification factor for PICO is 2.1. Please double-check.

Thank you for spotting this inconsistency, we have corrected this value and one more rounding error in the text.

L 187: As before, the authors may also add here the factor by writing: "... the lowest deep amplification (1.7), which is a ... ."

Agreed and adopted

L 193: Should it be: "These sensitivities have a factor between 1.7 (PICO) and 5.5 (Quadratic) higher than the ice-shelf average values."?

No, the sentence in the manuscript is correct. However, we reformulated to clarify what we mean by "deep sensitivities" as follows: *"These sensitivities in the deep regions are a factor 1.7 (PICO) to 5.5 (Quadratic) higher than the ice-shelf average values."*

L 226: The authors may rephrase "Quantitatively, however, the intermodel spread is very large" to "Quantitatively, however, the intermodel spread is considerably large."

Agreed and adopted

L 279.: The authors may add a sentence summarising the results for 2 °C: "For 2 °C, the increases range from xx % to xx %, with a mean of xx %."

Agreed and adopted